# Measuring and directing charge transfer in heterogenous catalysts

Michael J. Zachman [1][✉], Victor Fung[1,2], Felipe Polo-Garzon [3], Shaohong Cao[1], Jisue Moon [3], Zhennan Huang [1], De-en Jiang [2], Zili Wu [3] & Miaofang Chi [1][✉]

Precise control of charge transfer between catalyst nanoparticles and supports presents a unique opportunity to enhance the stability, activity, and selectivity of heterogeneous catalysts. While charge transfer is tunable using the atomic structure and chemistry of the catalyst-support interface, direct experimental evidence is missing for three-dimensional catalyst nanoparticles, primarily due to the lack of a high-resolution method that can probe and correlate both the charge distribution and atomic structure of catalyst/support interfaces in these structures. We demonstrate a robust scanning transmission electron microscopy (STEM) method that simultaneously visualizes the atomic-scale structure and sub-nanometer-scale charge distribution in heterogeneous catalysts using a model Au-catalyst/$SrTiO_3$-support system. Using this method, we further reveal the atomic-scale mechanisms responsible for the highly active perimeter sites and demonstrate that the charge transfer behavior can be readily controlled using post-synthesis treatments. This methodology provides a blueprint for better understanding the role of charge transfer in catalyst stability and performance and facilitates the future development of highly active advanced catalysts.

[1] Center for Nanophase Materials Sciences, Oak Ridge National Laboratory, Oak Ridge, TN 37831, USA. [2] Department of Chemistry, University of California, Riverside, CA 92521, USA. [3] Chemical Sciences Division, Oak Ridge National Laboratory, Oak Ridge, TN 37831, USA. [✉]email: zachmanmj@ornl.gov; chim@ornl.gov

Metal-support interactions significantly affect the reactivity, selectivity, and stability of oxide-supported metal nanocatalysts and therefore represent effective tuning parameters for optimizing catalyst performance[1–11]. Many of these interactions, including small-cluster stabilization, support participation in catalysis, and oxide encapsulation of metal nanoparticles[6,9,12], can be routinely accessed by various high-resolution microscopy techniques[9,11,13–15]. Directly studying charge transfer is more challenging, however. State-of-the-art theoretical calculations provide a powerful way to predict charge transfer and catalytic reaction mechanisms at the individual particle-level when representative structural models are given, but thus far experimental charge transfer studies have relied mainly on indirect or bulk experimental measurements[11,16,17]. Such methods cannot resolve the charge distribution around individual nanoparticles, and information about the influence of specific particle-support interface atomic configurations on charge redistribution is therefore difficult to obtain[12]. Scanned probe microscopy techniques have been used to retrieve information about local work function, and thus charge state, at high spatial resolution, but characterization of three-dimensional (3D) particles is challenging for these techniques, especially at particle perimeters, so these studies are often involve individual adatoms or two-dimensional "raft-like" particles on flat substrates[17–22]. Our understanding of charge transfer processes in real heterogenous catalysts therefore remains limited by a lack of techniques capable of directly studying these phenomena at their inherent nanometer length scales in individual 3D nanoparticles[23–25].

For example, heterogeneous catalysts composed of Au nanoparticles on oxide supports possess intriguing catalytic properties (e.g., CO oxidation at ambient conditions) and charge transfer between the Au nanoparticle and oxide support is believed to be critical to the catalytic behavior of these systems[1,26–30]. The exact role charge transfer plays in dictating this behavior, however, such as whether it is involved in the high activity of perimeter sites[31–33], is not well understood. For instance, it has been reported that negatively charged Au nanoparticles, i.e., particles that have gained electrons from the support, promote adsorption of CO molecules and weaken $O_2$ bonds[30], resulting in enhanced CO oxidation activity[34]. Cationic Au species, on the other hand, have been shown to have increased stability, a reduced activation barrier, and increased activity for CO oxidation[31,35]. Nevertheless, it is accepted that the structure and chemistry of the particle-support interface plays an important role in dictating transfer of charge[11,35–39]. The ability to provide direct experimental evidence of charge transfer features on the nanometer scale and correlate this with atomic-scale interfacial structure

would therefore significantly aid in resolving these ambiguities and enhance our understanding of charge transfer processes, allowing improved catalysts to be designed.

Here, using Au nanoparticles on a $SrTiO_3$ (STO) support as a model heterogeneous catalyst system, we probe charge redistribution around individual nanoparticles at the nanometer scale for the first time. To accomplish this, we adopt and further develop four-dimensional (4D) scanning transmission electron microscopy (STEM) techniques designed to map features related to projected electric and magnetic fields mainly in thin single crystals[40–51]. Using these techniques, we provide direct evidence that charge transfer occurs in the pristine Au-STO system from the support to the Au particle and that the overall direction of charge transfer can be inverted by a simple treatment aimed at modifying the support surface, which leads to altered catalytic activity. In addition, by pairing our experimental results with density functional theory (DFT) calculations, we provide new insights into the mechanisms behind the highly active perimeter sites of heterogeneous catalysts, which involve correlated charge transfer and local structural modifications.

## Results

**Scanning transmission electron microscopy**. Figure 1 shows representative atomic-resolution high-angle annular dark-field (HAADF) STEM images of Au nanoparticles on a (001)-oriented STO support. The atomic lattice of the Au particle is visible in Fig. 1b and the Bragg reflections in the corresponding fast Fourier transform (FFT) in Fig. 1c reveal that the particle is oriented with the <111> direction normal to the STO surface, a known stable orientation of Au on oxide supports[52,53].

While conventional STEM techniques provide valuable atomic-scale information, they are not typically sensitive to properties related to extended electrostatic fields. We therefore additionally employed 4D-STEM, which can be used to extract information related to internal electric fields and corresponding potentials and charge densities[40–48,50,51,54]. The center of mass (CoM) of the electron beam intensity in the detector plane, for example, measures momentum transfer from the specimen to the electron probe, and hence carries information about internal fields[45,46]. The same 4D dataset also allows us to reconstruct simultaneously acquired conventional STEM images such as annular dark-field (ADF) and bright-field (BF), which can provide structural and chemical information. Simultaneously acquired ADF- and BF-STEM images and an inverted CoM map taken of a Au nanoparticle on the STO support are shown in Fig. 2a–c. In addition to the background of atomic-scale contrast from the

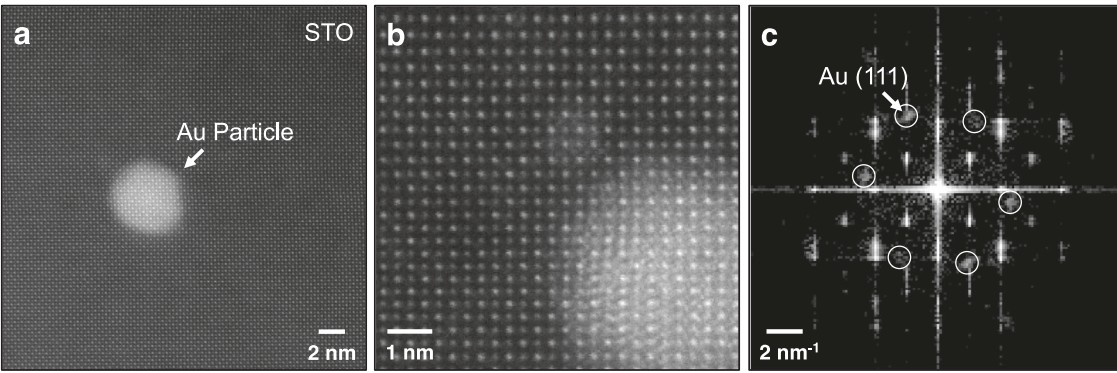

**Fig. 1 Atomic-scale structure of a heterogenous catalyst comprising Au nanoparticles on an (001)-oriented STO support, revealed by HAADF-STEM.**
**a** Low-magnification image of isolated Au nanoparticle on an STO support. **b** High-magnification image of the Au nanoparticle and **c** the corresponding FFT, with spots originating from the Au particle circled, showing the particle's orientation.

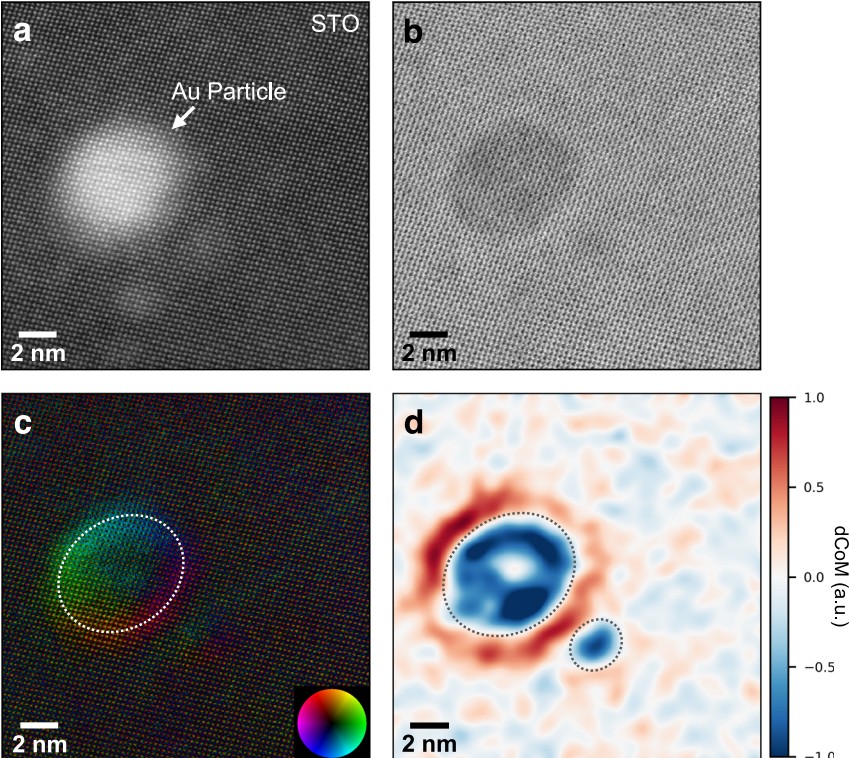

**Fig. 2 Atomic-scale STEM images and inverted CoM map acquired simultaneously in a 4D-STEM dataset, as well as filtered inverted dCoM map, of Au nanoparticle on STO. a** ADF-STEM and **b** BF-STEM images reconstructed from the 4D data. **c** Atomic-scale CoM map with direction and strength indicated by color and intensity, respectively (direction inverted from raw CoM to display features associated with electric fields appropriately, since the beam electrons are negatively charged). **d** Inverted dCoM map after application of a 4 Å Gaussian filter to isolate nanometer-scale features from the underlying atomic-scale information (see Supplementary Fig. 1 for a comparison with the original atomic-resolution map). The particles appear negative in projection, with the surrounding support positive. Dashed lines represent the approximate particle perimeters.

support, the inverted CoM map (Fig. 2c) reveals an extended feature in the region of the nanoparticle, reaching ~1–2 nm beyond its edge, indicating the possibility of an extended field.

Gauss's law allows information about the projected charge density in the specimen to be generated from information directly related to projected electric fields[45,46]. This provides the possibility for charge distributions, and thus the charge transfer behavior of metal-oxide supported metal nanoparticle catalysts, to be studied by 4D-STEM. Raw maps of the inverted divergence of the CoM shifts (dCoM) for Au-STO samples are shown in Supplementary Fig. 1, the contrasts of which are dominated by atomic-scale features originating from interactions of the beam with the STO atomic column potentials. Charge redistribution between the Au nanoparticle and support, however, would result in an electrostatic field that is relatively long-range and weak compared to atomic-scale fields, making direct detection of such features difficult in dCoM maps.

The most straightforward method for isolating long-range features is averaging over unit cells (Supplementary Fig. 2)[48], which was used as a first attempt at extracting information about any nanometer-scale charge distribution present. Figure 2d shows the inverted CoM map after application of a 4 Å Gaussian filter, comparable to the STO support lattice spacing, to minimize the contribution of the support lattice while preserving longer-range information (see Methods section). The effectiveness and appropriateness of using this simple CoM filtering method was validated by comparing results with those of a central disk shift tracking method designed to more directly extract long-range information[49], as discussed in detail in the Supplementary

Information. The projected charge density map in Fig. 2d shows that the Au nanoparticle is negatively charged. A positive region extends laterally from the particle edge ~2 nm into the STO support, as would be expected to compensate for transfer of negative charge to the particle, and which could itself alter catalytic activity[55]. In addition, the charge state near the perimeter region is especially important because it is believed that this is where critical reaction steps take place. In the CO oxidation reaction, for example, it is hypothesized that gold surfaces serve as CO adsorption sites and activation of oxygen molecules occurs at the particle perimeter[32,56]. Charge redistribution near the perimeter is therefore expected to influence oxygen activation and charge variation on the catalyst surface likely impacts transport of CO to the perimeter, both of which would affect the rate of oxidation reactions occurring there. In addition to the particle observed, a nearby smaller particle ~2 nm across (also circled in Fig. 2d) is just visible in the CoM map of Fig. 2c as well and can be seen more prominently in Fig. 2d. This particle is also negatively charged and appears to have a positively charged region surrounding it, similar to the large particle. This indicates the charge transfer direction is independent of particle size and also demonstrates that our method is capable of revealing charge redistribution effects in particles down to at least ~1 nm in radius.

**Density functional theory calculations.** To gain further insights into the origins of the features observed by 4D-STEM, we turned to DFT calculations of our Au-STO system. Calculations were

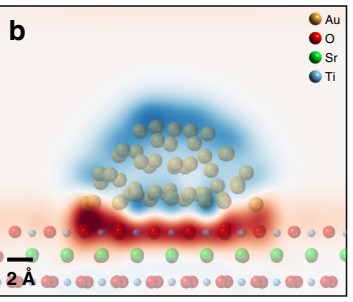
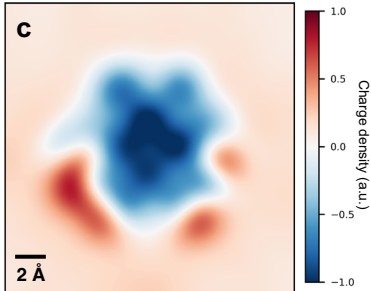

**Fig. 3 Charge transfer at a (111)-oriented Au nanoparticle/(001)-oriented STO support interface, calculated by DFT. a** Localized regions with high charge transfer are present under a pristine Au particle on an STO support, with Au-O bonds resulting in highly positive features. Data is displayed down the (110) STO zone axis. **b** The net charge on the particle is negative (blue) and the support is positive (red), as can be seen by averaging over the atomic-scale features of the charge transfer. **c** A top-down view reveals that, as in the experiments, the particle appears negative in projection, and the surrounding support positive.

performed by replicating the structural configuration observed in our experiments, i.e., a Au nanoparticle oriented with the <111> direction normal to a (001) STO surface (see Methods section for details). Systems with both SrO- and $TiO_2$-terminated STO surfaces were calculated, and the results showed that termination does not significantly affect the form of the charge redistribution in this case. Additional discussions can be found in the Supplementary Information and Supplementary Fig. 3. As such, we discuss here only the $TiO_2$-terminated case, though the discussion should generally apply to either surface termination. The relaxed catalyst structure that results from the calculations is shown in Supplementary Fig. 4, and Fig. 3a, b shows the corresponding calculated charge transfer, defined as the difference between the charge distribution of the combined nanoparticle-support system and that of the sum of the isolated nanoparticle and support, with the atomic structure overlaid. The calculations predict that charge transfer occurs from the STO support to the Au nanoparticle, resulting in an overall negative charge on the particle and a positive support. While this charge transfer occurs largely at the nanoparticle-support interface, the top-down view (the configuration observed in our experiments) in Fig. 3c shows that the particle remains negative in projection, with the surrounding STO positive, in agreement with the experimental 4D-STEM results.

As opposed to the net negative charge on the particle, individual Au atoms at the particle-support interface can become positively charged when in close proximity to support O atoms (Fig. 3a). This leads to localized highly positive regions at the particle perimeter in projection (Fig. 3c). Simultaneously, the local structure of the interface is strongly reconstructed at these atoms, resulting in decreased Au-O distances and increased Au-Au distances. These structural distortions appear amplified at the particle perimeter as well, where the lower coordination of Au atoms renders them more prone to displacement. The charge state and local structural environment of these perimeter atoms therefore varies significantly from other surface atoms on the particle, suggesting they may be related to the known high activity of perimeter sites.

**Reversing the charge transfer direction**. Precise control over the direction of charge transfer at catalyst-support interfaces is critical for the design and optimization of heterogenous catalysts. Different charge schemes for Au nanoparticles, i.e., positively or negatively charged, are favored for different catalytic reactions[57–59]. Here, we provide direct evidence that inverting the charge transfer direction is feasible using post-synthesis treatments. As described in the Methods section, the specimen shown in Figs. 1 and 2 was treated in hydrogen gas after synthesis to ensure no excess oxygen was present. To explore the possibility of

altering the charge transfer features of this system, we added a subsequent heat treatment step in oxygen gas at 300 °C for 4 h to intentionally introduce additional oxygen into the system. For full details of the procedure, see the Methods section. We refer to the initial sample and the sample subsequently treated in oxygen as "H-treated" and "O-treated," respectively. To demonstrate the effect of the oxygen treatment, we again utilized the 4D-STEM method introduced above to image the post-treatment charge distribution.

Figure 4 shows a HAADF image and 4D-STEM inverted dCoM result for the O-treated catalyst, filtered as in Fig. 2d (see original map in Supplementary Fig. 1c). Remarkably, the results show that the overall charge transfer direction for the O-treated catalyst is opposite that of the H-treated catalyst, resulting in positively charged particles. Besides the overall positive charge on the particles, the charge appears to be localized near the perimeter of the particles, which again may help explain why the perimeter sites often serve as the active sites in reactions. As expected, these positive particles are surrounded by a negative region in the STO. Further, a larger magnitude of negative is observed between the two closely neighboring particles in the upper-left corner of Fig. 4, suggesting that spatial assembly of nanoparticles may also serve as a means for tuning charge distributions of catalysts and therefore their catalytic performance[55]. In addition, the opposite charge states of Au measured in this study further validate the CoM measurements of charge distribution. The potential impact of secondary electron emission and plasmonic excitations is minimal (Supplementary Discussions).

DFT calculations emulating the O-treated case provide theoretical support for the positive charge on Au particle as well (structural models are discussed in the Supplementary Information and the used relaxed structure is shown in Supplementary Fig. 4). As shown in Fig. 5, we observe that oxygen atoms at the perimeter of the Au particle produce charge transfer features consistent with our experimental 4D-STEM results. In addition to inversion of the overall charge transfer direction, three other distinct differences are present between the O-treated and H-treated catalysts. First, in the O-treated case, the positive charge on the particle is largely present near its perimeter, while the negative charge is more homogeneously spread across the particle in the H-treated case. Second, the charge distribution around the perimeter is more symmetrical for the O-treated case (Fig. 5c) than the H-treated case (Fig. 3c). Finally, distortions of the Au lattice at the perimeter of the O-treated particle are significantly reduced compared to the H-treated case. Since charge states and lattice properties are correlated and affect the strength of metal-adsorbate bonding[60,61], the distinctive charge distributions and local perimeter structures introduced by the two treatments likely result in different active sites, altered transport

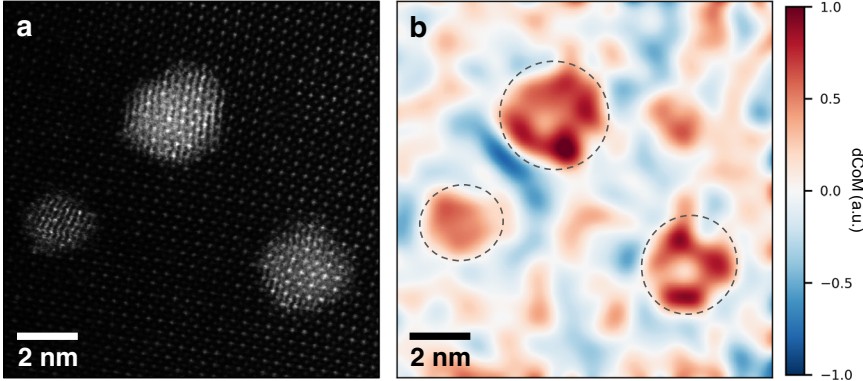

**Fig. 4 Effect of O-treatment on charge transfer as measured by dCoM 4D-STEM. a** HAADF image of an O-treated Au-STO catalyst taken before the 4D-STEM map. **b** Charge density map of the catalyst particles after application of a 4 Å Gaussian filter to the original atomic-resolution inverted dCoM map (shown in Supplementary Fig. 1). In the O-treated case, the particles appear positive in projection, with a negative region in the surrounding support. Much of the positive charge appears near the perimeter of the particles.

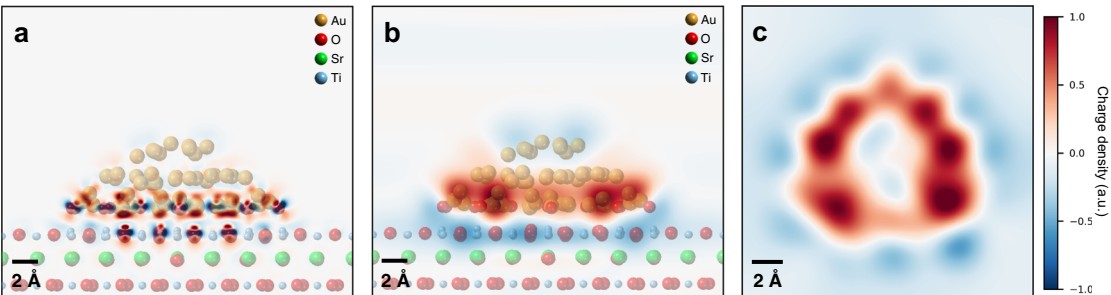

**Fig. 5 Charge transfer at a (111)-oriented Au nanoparticle/(001)-oriented STO support interface, with additional oxygen at the particle perimeter, calculated by DFT. a** Additional oxygen atoms around the perimeter of the particle, emulating the O-treated case, significantly alter the charge transfer features of the system in Fig. 3. **b** The sign of the net charge on the particle and support is inverted, with the particle positive and the support negative. **c** In top-down projection, the positive features appear mainly localized to the particle perimeter, consistent with the experimental results.

mechanisms of adsorbed molecules, and a modification of the delicate balance between elementary steps in catalytic reactions. It should also be pointed out that these results demonstrate the intricacy of the nano-environment of heterogenous catalyst interfaces. Minute changes to the system, such as particle size, interfacial facets, vacancies/dopants, and impurities are thus expected to modify the charge distribution and lattice strain/defects of a catalyst, and hence its performance. Methods such as those described here are therefore necessary for exploring the consequences of such modifications.

**CO oxidation reaction case study**. To explore the impact of charge transfer on catalytic performance, we used CO oxidation performance of our H- and O-treated Au-STO catalysts as a case study. A commercial STO nanoparticle support with an increased Au loading were used (see Methods and Supplementary Fig. 5 for sample details). As expected, the two catalysts showed different catalytic performances. As shown in Fig. 6, a higher conversion rate was found for the H-treated Au-STO catalyst than the O-treated sample, while the pure STO support showed little conversion with either treatment. These results suggest that negatively charged Au catalysts generally provide better CO oxidation performance than positively charged Au nanoparticles. This is consistent with differences in carbon monoxide and oxygen absorption strengths, for example, which are thought to be enhanced on negatively charged Au particles

and suppressed for $O_2$ molecules on positively charged Au particles[62]. However, as mentioned earlier, previous reports on mechanisms connecting charge state to catalytic performance have not been consistent.

Rather than a simple dependence on the overall charge transfer direction, the results presented here suggest that the specific features of the particle and support charge distribution, as well as the structural and chemical configuration of the perimeter region, play a large role in defining active sites and governing reaction performance. For example, we found that the charge on Au atoms near support O atoms can be positively charged despite an overall net negative charge on the particle. Such an evident spatial variation in charge transfer coincides with a considerable distortion of the local Au lattice, and could further impact reaction pathways. In addition, extra O atoms near the perimeter suppress these local positive charges and the corresponding structural distortions while inducing an overall positive charge on the particle. If reactions occur primarily at the locally positively charged perimeter sites of the net negatively charged H-treated particle, suppression of these sites in the overall positively charged O-treated case may explain its decreased performance. The resulting disparity between the local active site charge state and overall net particle charge may have contributed to complication of previous attempts to directly relate charge state to activity. Despite the increased understanding provided here, the complete reaction mechanisms remain complicated and merit further detailed theoretical investigation. These results do suggest,

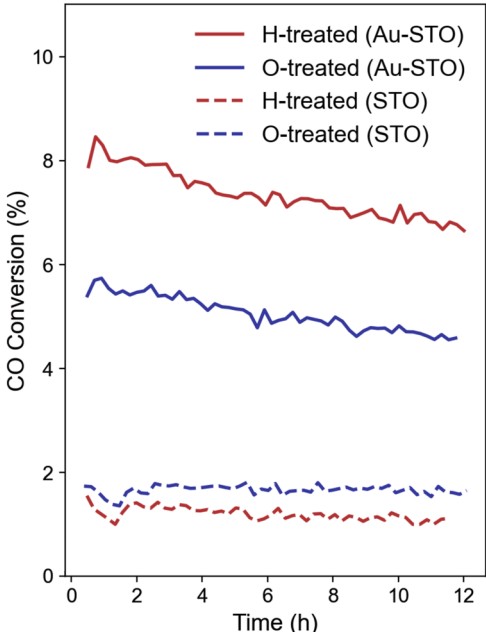

**Fig. 6 Effect of catalyst charge state on performance.** H-treated catalyst has increased CO conversion performance over O-treated catalyst (solid lines). Dashed lines indicate the performance of the STO substrate without Au catalyst particles, treated with the same procedures as the Au-STO catalysts.

however, that simply tuning the overall net charge on a particle may not be sufficient to optimize its catalytic properties, and properly coupled structural and interfacial properties designed to modify the specific local structure and charge of perimeter sites may also be necessary. In the future, correlated atomic-scale structural configurations and charge distributions provided by 4D-STEM will serve as reliable experimental input for theory and allow reaction mechanisms of real catalysts with different charge states to be revealed.

## Conclusion

The advanced 4D-STEM techniques demonstrated here enable charge transfer to be studied at the level of individual catalyst-support pairs with spatial resolution sufficient to provide direct input for theoretical calculations. Combined with conventional STEM images simultaneously acquired and those reconstructed from the same 4D datasets, these techniques enable direct correlation between atomic structure and charge distribution in heterogeneous catalysts, which was previously challenging. By applying these techniques to an STO-supported Au catalyst model system we directly mapped charge redistribution at the nanometer scale in a heterogeneous catalyst for the first time, demonstrated that the charge transfer direction between the metal and support can be reversed by post-synthesis treatments, and by pairing our results with DFT calculations we revealed the presence of coupled local charge redistribution and structural distortions at the particle-support perimeter. These findings may be vital for explaining the localization of active sites at the perimeter regions of metal-oxide–supported metal nanocatalysts and help to resolve previous ambiguities about the relationship between charge state and performance. In the future, these techniques will enable the detailed effects of different structural and chemical configurations, e.g., various particle sizes, support species, and surface facets, on charge transfer to be studied. Integration with state-of-the-art in situ microscopy methods will allow direct

visualization of dynamic changes in catalyst charge state during reactions, further accelerating the design of high-performance catalysts.

## Methods

**Electron microscopy data acquisition and analysis.** A JEOL NEOARM aberration-corrected STEM was used to collect the 4D datasets. The instrument was operated at 200 kV with a semiconvergence angle of 28 mrad for the main text results, resulting in an Å-scale probe, and 7 mrad and 2.5 mrad semiconvergence angles for the additional results shown in Supplementary Fig. 7, resulting diffraction-limited probe sizes of approximately 3.6 Å and 1 nm, respectively. The 4D data was acquired on a PNDetector pnCCD (S)TEM Camera with 0.5 or 1 ms/pixel dwell times and a recorded resolution of $264 \times 132$ or $264 \times 264$ pixels, respectively, and data was acquired over an array of up to $512 \times 512$ probe positions.

Common Python packages such as NumPy, SciPy, and Matplotlib were used to analyze and display the data. The CoM was calculated using data from the center to outside of the central diffraction disk. The relative rotation between the scan and detector orientations was accounted for using the angle observed between the real space lattice and the support diffraction pattern. From these measurements, the divergence of the inverted CoM was then calculated to reveal features associated with the long-range charge density after subsequent application of a Gaussian filter. Gaussian filters such as this locally weight intensity values around a given point using a two-dimensional Gaussian function and average them to minimize high spatial-frequency information while preserving low-frequency information. Due to the nature of the Gaussian function, this low-pass filter method does not produce artifacts such as high-frequency ringing[63]. Further, more robust methods for extracting the charge density are discussed and compared with this method in the Supplementary Information. Particle size measurements were performed by taking line profiles across particles and measuring their diameter.

**Sample synthesis and preparation.** To synthesize the samples, single-crystal STO STEM specimens were first prepared by mechanical polishing, followed by ion milling. The thin region of the resulting specimens is ~10 nm thick. Comparison of simulated and experimental data, discussed below, confirms that the STO is on the order of 10 nm thick for the sample in Fig. 3a and <5 nm for that in Fig. 3b. The incipient wetness impregnation method was then used to deposit Au nanoparticles on the STO sample, as reported previously[64]. Briefly, an aqueous solution of 17 mg $HAuCl_4 \cdot 3H_2O$ in 20 mL of deionized water was prepared, and 3 µL of the solution was impregnated on each sample. The samples were dried overnight in a vacuum oven at room temperature and were then treated at 300 °C for 4 h in 35 mL/min of 4% $H_2$/Ar, which we designate here the "H-treated" sample, or for 4 h in 35 mL/min of 4% $H_2$/Ar at 300 °C followed by 4 h in 35 mL/min of 10% $O_2$/Ar at 300 °C, which we designate here the "O-treated" sample. Powder Au/STO samples were synthesized via incipient wetness impregnation as well. $HAuCl_4 \cdot 3H_2O$ was dissolved in 0.5 mL of deionized water and mixed with commercial $SrTiO_3$ (Sigma-Aldrich). Inductively coupled plasma (ICP) spectroscopy was performed by Galbraith Laboratories Inc. The loading of Au in the sample was measured to be 0.15 wt.%. Same treatments as those for the Au/single crystalline STO were performed for the powder samples.

**Density functional theory calculations.** The DFT calculations were performed using the Vienna Ab Initio Simulation Package (VASP)[65,66]. The Perdew-Burke-Ernzerhof (PBE)[67] functional form of generalized-gradient approximation (GGA) was used for electron exchange and correlation energies. The projector-augmented wave method was used to describe the electron-core interaction[65,68]. A kinetic energy cutoff of 450 eV was used for the plane waves. The Brillouin zone was sampled at the gamma point only. A vacuum layer of 15 Å was added for the surface slabs along the z-direction. The $SrTiO_3$ slab contains a total of four layers, with the bottom two layers fixed in their bulk-optimized positions from DFT, and with the top two layers and Au atoms allowed to move. The Au cluster in Fig. 4a–c consisted of 52 atoms cut from a pristine bulk Au lattice, oriented with the (111) direction normal to the STO surface before relaxation. The cluster size was chosen to be large enough to be within the range of experimentally observed particle sizes (~1.4 Å) while remaining computationally tractable for DFT calculations. A semi-spherical cluster was used as the initial state for the relaxation to mimic common morphologies observed for Au supported on oxides[69]. To emulate the O-treated case, oxygen atoms were added to the interfacial regions of the cluster on the bridging sites between Au and surface Ti and locally optimized with DFT. These interfacial oxygen atoms were placed in the most stable locations according to our own studies, and these are consistent with previous computational work that found the bridging Ti-O-Au to be the most stable for Au cluster–TiO2 interfacial sites[58,70]. The data in Fig. 4a, b, d, e and c, f were displayed in projection down the (110) and (001) STO directions, respectively.

## Data availability
The data that support the findings of this study are available from the corresponding authors upon reasonable request.

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

## Acknowledgements

Research was sponsored by the US DOE, Office of Science, Office of Basic Energy Sciences (BES), Chemical Sciences, Geosciences, and Biosciences Division, Catalysis Science program. Technique development and data analysis were supported by US DOE Office of Science under Early Career award no. ERKCZ55. All microscopy research was supported by the Center for Nanophase Materials Sciences (CNMS), which is a US Department of Energy, Office of Science User Facility at Oak Ridge National Laboratory. We would like to thank Nina Balke for useful conversations. This research used resources of the National Energy Research Scientific Computing Center, a DOE Office of Science User Facility supported under contract no. DE-AC02-05CH11231.

## Author contributions

M.C. conceived the research, prepared the STEM specimens, acquired the high-resolution STEM images, and contributed to data interpretation and writing of the manuscript. M.J.Z. designed aspects of the 4D-STEM experiments, acquired the O-treated sample 4D-STEM data sets, analyzed and interpreted the 4D-STEM data, visualized the other types of data, and prepared the manuscript. V.F. and D.J. performed the DFT calculations and contributed to data interpretation. S.C. acquired the H-treated sample 4D-STEM data sets. J.M., F.P.G., and Z.W. synthesized the samples, conducted the performance measurements, and contributed to data interpretation. Z.H. collected STEM images and quantified particle sizes. All authors contributed to revising the manuscript.

## Competing interests

The authors declare no competing interests.
