## [Peer Review File · Nature Communications]

Title: Measuring and directing charge transfer in heterogenous catalystsREVIEWER COMMENTS

Reviewer #1 (Remarks to the Author):

In this manuscript, the authors demonstrate an advanced method of CoM map to probe the charge distribution of catalyst/support interfaces. The charge state of supported metals is one of the most significant aspects that determine the catalytic performance, which have drawn tremendous attention in recent years. As mentioned in the manuscript, it's challenging to resolve the charge distribution around individual three-dimensional (3D) nanoparticles. With the mentioned methodology, the authors can experimentally reveal that a charge transfer occurs from the STO support to the Au particle in the pristine Au-STO system, and the overall direction of the charge transfer can be inverted by modification of the support surface. Density functional theory calculations also support the charge analysis. This work provides a robust case study of CO oxidation to explore the impact of charge transfer on the catalytic performance and proves that negatively charged Au catalysts possess a better performance. Hence, I would recommend its publication in Nature communications after major revision.

Some concerned issues are listed below:

- 1) It is an important result of the CO oxidation case study to explore the impact of the charge transfer on catalytic performance. The supplementary Fig. 6 is better to be moved to the main text. Also a previous relating paper (JACS 2018, 140, 554) should be cited.
- 2) The claim in the abstract "While charge transfer is tunable using the atomic structure and chemistry of the catalyst-support interface, direct experimental proof is missing, primarily due to the lack of a high-resolution method that can probe and correlate both charge distribution and atomic structure of catalyst/support interfaces." is arbitrary and baseless. Please refer to a recent on-line paper (J. Mater. Chem. A, DOI: 10.1039/d1ta08353h) and references therein, which clearly show that SPM (STM and AFM) can obviously provide the information on both the charge distribution and atomic structure of catalyst/support interfaces. These references should be cited.
- 3) It's not precise to use "sample potential" to describe the SPM method to measure the charge state in line 46 of page 3. Instead, "local work function" or "potential barrier" is widely used in literature like Angew. Chem. Int. Ed. 2020, 59, 14321 and Nat. Mater. 2010, 9, 320.
- 4) It's unclear why a 4 Å Gaussian filter should be applied to show an obvious change from the original dCoM map (Fig. S1). Are there any possible artificial effects that may be introduced during this data processing? The authors should elaborate more on this.
- 5) Refs. 39, 47, 48 and 52 lack of detailed information.

Reviewer #2 (Remarks to the Author):

The manuscript deals with a very important problem of identifying, tuning and understanding electron transfer taking place between deposited metal particles and the underlying supports, the systems constituting typical heterogeneous catalysts. The authors successfully demonstrate capabilities of a STEM method to simultaneously visualize the atomic structure and the charge distribution in it that opens new opportunities to control the charge transfer behavior. The applied experimental approach

allowing to qualitatively determining charge transfer will be helpful for clarifying the longstanding puzzles of relationships between the charge state of catalytically active sites and their performance. This clearly written manuscript can be recommended for publication after a minor revision addressing the following issues:

1) The STEM visualization qualitatively characterizes the charge distribution. A comment on the possibility of quantitative charge distribution measurements using the same experimental setup would be helpful.

2) The bottom two layers of the four-layer SrTiO₃ slab were “fixed in their bulk positions” (p. 26, line 553) in the DFT calculations. Please clarify, whether the experimental or DFT-optimized bulk positions were used.

3) A more detail justification of using Au₅₂ model cluster cut from the bulk lattice of gold is due. How its size and shape are related with those of the experimentally studied samples? Also, the structure resulting from the local geometry optimization of such bulk-cut Au₅₂ model cluster may be quite different from the lowest-energy structure, thus causing notably overestimated reactivity of the chosen model. Finally, in the O-treated models, were the added interfacial O atoms in the most stable calculated positions or there are other, more stable positions for these O atoms?

March 21, 2022

Re: Decision on Nature Communications manuscript NCOMMS-22-01757-T

Dear Referees and Editor,

Thank you very much for your constructive comments. We have responded to your comments below and edited our manuscript accordingly, with significant changes outlined below. Please also find attached two versions of the revised manuscript, one with and one without changes highlighted.

Reviewer comments:

Reviewer #1:

In this manuscript, the authors demonstrate an advanced method of CoM map to probe the charge distribution of catalyst/support interfaces. The charge state of supported metals is one of the most significant aspects that determine the catalytic performance, which have drawn tremendous attention in recent years. As mentioned in the manuscript, it's challenging to resolve the charge distribution around individual three-dimensional (3D) nanoparticles. With the mentioned methodology, the authors can experimentally reveal that a charge transfer occurs from the STO support to the Au particle in the pristine Au-STO system, and the overall direction of the charge transfer can be inverted by modification of the support surface. Density functional theory calculations also support the charge analysis. This work provides a robust case study of CO oxidation to explore the impact of charge transfer on the catalytic performance and proves that negatively charged Au catalysts possess a better performance. Hence, I would recommend its publication in Nature communications after major revision. Some concerned issues are listed below:

We appreciate the reviewer's assessment of the importance of our work and for their valuable suggestions, which we have addressed below.

1) It is an important result of the CO oxidation case study to explore the impact of the charge transfer on catalytic performance. The supplementary Fig. 6 is better to be moved to the main text. Also a previous relating paper (JACS 2018, 140, 554) should be cited.

We are grateful that the reviewer finds the CO oxidation data important and have moved it to the main text as Fig. 6, as suggested. The recommended reference was included where SPM techniques are discussed in the introduction:

*"... these studies are often **involve** individual adatoms or two-dimensional "raft-like" particles on flat substrates¹⁷⁻²²."*

2) The claim in the abstract "While charge transfer is tunable using the atomic structure and chemistry of the catalyst-support interface, direct experimental proof is missing, primarily due to the lack of a high-resolution method that can probe and correlate both charge distribution and atomic structure of catalyst/support interfaces." is arbitrary and baseless. Please refer to a recent on-line paper (J. Mater. Chem. A, DOI: 10.1039/d1ta08353h) and references therein, which clearly show that SPM (STM and AFM) can obviously provide the information on both the charge distribution and atomic structure of catalyst/support interfaces. These references should be cited.

We thank the reviewer for pointing out that this was stated too broadly. As discussed in the introduction, SPM techniques are capable of measuring the charge state of individual atoms and two-dimensional catalyst particles at high resolution. The technique described here extends these capabilities to three-dimensional nanoparticles on supports. We have therefore clarified this statement to make this distinction more clear:

*“While charge transfer is tunable using the atomic structure and chemistry of the catalyst-support interface, direct experimental **evidence** is missing for **three-dimensional catalyst nanoparticles**, primarily due to the lack of a high-resolution method that can probe and correlate both **the** charge distribution and atomic structure of catalyst/support interfaces **in these structures**.”*

In addition, the paper and selected references within are now cited in the introduction (see response to previous comment).

3) It's not precise to use “sample potential” to describe the SPM method to measure the charge state in line 46 of page 3. Instead, “local work function” or “potential barrier” is widely used in literature like Angew. Chem. Int. Ed. 2020, 59, 14321 and Nat. Mater. 2010, 9, 320.

We appreciate the reviewer advising us on the most appropriate terms for SPM measurements. “Sample potential” has now been changed to “local work function” in the manuscript where these techniques are discussed, as the reviewer recommended:

*“Scanned probe microscopy techniques have been used to retrieve information about **local work function**, and thus charge state, at high spatial resolution ...”*

4) It's unclear why a 4 Å Gaussian filter should be applied to show an obvious change from the original dCoM map (Fig. S1). Are there any possible artificial effects that may be introduced during this data processing? The authors should elaborate more on this.

We would like to thank the reviewer for pointing out that the Gaussian filtering process may not be obvious to readers in fields where this procedure is not as commonly performed as it is in electron microscopy, and we should therefore further explain this process.

A Gaussian filter is most directly described as a weighted average of local values around a given point, with the weights being given by to a two-dimensional Gaussian function. Another way to visualize this is a convolution of the image with a two-dimensional Gaussian function. As a result, the Gaussian filter acts as a low-pass filter, removing high-frequency (small spatial size) information while preserving low-frequency (large spatial size) information [<https://www.sciencedirect.com/science/article/pii/B978012374457900010X?via%3Dihub>].

In this manuscript, a Gaussian function with a 4 Å width was used since the underlying STO lattice has a spacing of approximately 4 Å. As a result, information from an area approximately equal to a unit cell was used to produce the image at a given point, after Gaussian filtering. Signals with special frequencies approximately equal to or higher than the atomic spacing of the lattice were therefore averaged over, while signals with lower frequencies (such as the nm-scale charge transfer features) remained minimally affected. In addition, a Gaussian filter is a true low-pass filter, meaning that it preserves low-frequency information without producing high-frequency artifacts [<https://www.sciencedirect.com/science/article/pii/B978012374457900010X?via%3Dihub>]. That being said, edges of images must be handled with care, since a weighted average of the local region is complicated near edges by the lack of data outside of the image. As a result, we set the area outside of the image equal to zero for our processing, since the net charge within the image should be approximately zero. While this minimizes spurious contrast near the edges, regions approximately the width of the filter were cropped so that information in these areas was not displayed.

To further explain the details of the Gaussian filtering process, we have included additional text in the Methods section and in the main text to refer readers here:

*“Figure 2d shows the CoM map after application of a 4 Å Gaussian filter, comparable to the STO support lattice spacing, to minimize the contribution of the support lattice while preserving longer-range information (see **Methods**).”*

“Gaussian filters such as this locally weight intensity values around a given point using a two-dimensional Gaussian function and average them to minimize high spatial-frequency information while preserving low-

frequency information. Due to the nature of the Gaussian function, this low-pass filter method does not produce artifacts such as high-frequency ringing⁶³.

5) Refs. 39, 47, 48 and 52 lack of detailed information.

We appreciate the reviewer pointing this out, and the full reference details have now been added.

Reviewer #2:

The manuscript deals with a very important problem of identifying, tuning and understanding electron transfer taking place between deposited metal particles and the underlying supports, the systems constituting typical heterogeneous catalysts. The authors successfully demonstrate capabilities of a STEM method to simultaneously visualize the atomic structure and the charge distribution in it that opens new opportunities to control the charge transfer behavior. The applied experimental approach allowing to qualitatively determining charge transfer will be helpful for clarifying the longstanding puzzles of relationships between the charge state of catalytically active sites and their performance.

This clearly written manuscript can be recommended for publication after a minor revision addressing the following issues:

We are thankful for the reviewer's kind assessment of our work, and we agree that the technique has the potential to significantly impact the field of catalysis. Below, we have addressed the reviewer's comments.

1) The STEM visualization qualitatively characterizes the charge distribution. A comment on the possibility of quantitative charge distribution measurements using the same experimental setup would be helpful.

We would like to thank the reviewer for bringing up this important point. While it is possible to assign quantitative values to the charges displayed in the figures since beam deflections are directly related to the sample potential (and hence charge density), we do not feel it is instructive to do so at this stage so for the following reasons.

First, the strength of the displayed CoM shifts depends on the width of Gaussian filter used and is also a function of the spatial frequency of the features observed after filtering. As a result, quantitative values of charge measured would be a function of the chosen Gaussian filter size. While we chose the smallest filter size that removed the atomic-scale information, some effect on the observed strength of the nm-scale charge transfer features would be present. Similarly, choice of convergence angle can affect the measured charge value as well. As the probe size approaches the feature size (such as for the small convergence angle data shown in Supplementary Fig. 7), the strength of the detected features is reduced. For example, the maximum beam deflection observed in Supplementary Fig. 7 decreases from the 28 mrad to the 7 mrad to the 2.5 mrad convergence angle data. As a result, any quantitative values provided would be dependent on these variables.

In addition, changes in thickness of a material contributes to CoM/beam deflections, as shown in Supplementary Fig. 8. While we believe this contribution is not responsible for the majority of the qualitative features observed, it would not have a negligible contribution to a quantitative measurement, shifting observed charges on particles in the positive direction.

Finally, imaging through the catalyst and support in projection makes observation of the regions of strongest charge accumulation, at the particle-support interface, challenging. In the future, it may be possible to comment on these charges using a perpendicular sample orientation (such as the DFT results shown in Fig. 5a,b), but this also significantly increases the difficulty of data interpretation since the substrate is no longer uniform beneath the particle, resulting in significant changes in sample thickness across the field of view, which also cause beam deflections as mentioned above.

As a result, while it is possible to assign qualitative numbers to the charge transfer observed, we believe that the results should currently be presented primarily through a qualitative matching of experimental and theoretical

features, as performed in this manuscript. We have added a section about these points to the Supplementary Information:

“Quantitative data interpretation.

While it is possible to assign quantitative values to the charges observed, it is not instructive to do so at this stage. The strength of beam deflections observed is a function of the width of Gaussian filter used and the spatial frequency of the features present. While we chose the smallest filter size that could remove the atomic-scale CoM information in the main text, some effect on the strength of the observed nm-scale charge transfer features would be present. Similarly, choice of convergence angle can affect the magnitude of the measured charge. As probe size approaches the feature size, such as in Supplementary Fig. 7, the strength of observed beam deflections is reduced. As a result, quantitative charge values provided would be dependent on these variables. In addition, changes in thickness of a material contribute to beam deflections, as shown in Supplementary Fig. 8. While we believe this contribution is not responsible for the majority of the qualitative features observed here, the contribution to a quantitative measurement would be non-negligible, shifting observed particle charges in the positive direction. Finally, imaging in projection through the catalyst and support makes observation of the regions of strongest charge accumulation at the particle-support interface challenging. In the future, it may be possible to measure these charges using a perpendicular sample orientation (much like the DFT results shown in Fig. 5a,b), but this also significantly increases the difficulty of data interpretation since the substrate is no longer uniform beneath the particle, resulting in significant changes in sample thickness across the field of view and causing beam deflections as discussed above.

As a result, while it is possible to assign qualitative numbers to the charge transfer observed, we believe that results should currently be presented primarily through a qualitative matching of experimental and theoretical features, as performed in this manuscript.”

2) The bottom two layers of the four-layer SrTiO₃ slab were “fixed in their bulk positions” (p. 26, line 553) in the DFT calculations. Please clarify, whether the experimental or DFT-optimized bulk positions were used.

We appreciate the reviewer suggesting additional clarification here. The bulk-optimized DFT positions were used, and as such we have clarified this in the Methods:

“...the bottom two layers fixed in their bulk-optimized positions from DFT, and with the top two layers and Au atoms allowed to move...”

3) A more detail justification of using Au₅₂ model cluster cut from the bulk lattice of gold is due. How its size and shape are related with those of the experimentally studied samples? Also, the structure resulting from the local geometry optimization of such bulk-cut Au₅₂ model cluster may be quite different from the lowest-energy structure, thus causing notably overestimated reactivity of the chosen model. Finally, in the O-treated models, were the added interfacial O atoms in the most stable calculated positions or there are other, more stable positions for these O atoms?

We thank the reviewer for bringing up these important points. First, the cluster size was chosen large enough to be within the range of experimentally observed particle sizes (~1.4 angstroms) while remaining computationally tractable for the DFT calculations. A semi-spherical cluster was used as the initial state in the relaxation to mimic the common morphologies observed in supported Au on oxides [<https://www.pnas.org/doi/pdf/10.1073/pnas.1800262115>]. In addition, from previous experimental and theoretical studies, it was observed that clusters of ~2nm or lower were highly dynamic or fluxional [<https://www.pnas.org/doi/pdf/10.1073/pnas.1800262115>, <https://www.nature.com/articles/ncomms7511>], which does not suggest the existence of a stable global minimum state. The observations drawn from the computational modeling were also found to be general and not dependent on a particular conformation, however, especially as we tested clusters on different terminations and slightly different sizes and found similar charge transfer properties. As for the oxygen configuration, the interfacial oxygen atoms were placed in the most stable locations according to our own studies, which are consistent with previous computational work(s) as well, where the bridging Ti-O-Au were found to be the most stable for interfacial sites for Au clusters on TiO₂ [<https://pubs.acs.org/doi/10.1021/ja402063v>, <https://pubs.acs.org/doi/10.1021/jacs.6b04187>].

To clarify these points, additional details have been added to the Methods section:

*“The Au cluster in Fig. 4a-c consisted of 52 atoms cut from a pristine bulk Au lattice, oriented with the (111) direction normal to the STO surface before relaxation. **The cluster size was chosen to be large enough to be within the range of experimentally observed particle sizes (~1.4 Å) while remaining computationally tractable for DFT calculations. A semi-spherical cluster was used as the initial state for the relaxation to mimic common morphologies observed for Au supported on oxides⁶⁹.** To emulate the O-treated case, oxygen atoms were added to the interfacial regions of the cluster on the bridging sites between Au and surface Ti and locally optimized with DFT. **These interfacial oxygen atoms were placed in the most stable locations according to our own studies, and these are consistent with previous computational work that found the bridging Ti-O-Au to be the most stable for Au cluster-TiO₂ interfacial sites^{58,70}.**”*

We would once again like to express our gratitude to the reviewers for their constructive comments, and we feel that the additions and modifications made based on these comments have strengthened our manuscript. Please do not hesitate to contact us with any further comments or requests.

Sincerely,

Michael Zachman and Miaofang Chi
On behalf of all authors

REVIEWERS' COMMENTS

Reviewer #1 (Remarks to the Author):

In the revised version, the authors have addressed my previous concerns one by one and made changes accordingly. I have carefully read the revised text and would therefore recommend its acceptance for publication in Nature Communications.